

# A national landslide inventory of Denmark

Gregor Luetzenburg[1], Kristian Svennevig[2], Anders A. Bjørk[1], Marie Keiding[2], Aart Kroon[1]

[1]Department of Geosciences and Natural Resource Management, University of Copenhagen, Copenhagen, Denmark

[2]Geological Survey of Denmark and Greenland (GEUS), Copenhagen, Denmark

*Correspondence to*: Gregor Luetzenburg (gl@ign.ku.dk) and Kristian Svennevig (ksv@geus.dk)



**Abstract.**

Landslides are a frequent natural hazard occurring globally in regions with steep topography. Additionally, landslides are
playing an important role in landscape evolution by transporting sediment downslope. Landslide inventory mapping is a
common technique to assess the spatial distribution and extend of landslides in an area of interest. High-resolution digital
elevation models (DEMs) have proven to be useful databases to map landslides in large areas across different land covers and
topography. So far, Denmark had no national landslide inventory. Here we create the first comprehensive national landslide
inventory for Denmark derived from a 40 cm resolution DEM from 2015 supported by several 12.5 cm resolution orthophotos.
The landslide inventory is created based on a manual expert-based mapping approach, and we implemented a quality control
mechanism to assess the completeness of the inventory. Overall, we mapped 3202 landslide polygons in Denmark with a level
of completeness of 87%. The landslide inventory can act as a starting point for a more comprehensive hazard and risk reduction
framework for Denmark. Furthermore, machine-learning algorithms can use the dataset as a training dataset to improve future
automated mapping approaches. The complete landslide inventory is made freely available for download at
https://doi.org/10.6084/m9.figshare.16965439.v1   (Svennevig   and   Luetzenburg,   2021)   or   as   web   map
(https://data.geus.dk/landskred/) for further investigations.





## 1 Introduction

Landslides can be a serious natural hazard, existing worldwide causing high numbers of fatalities and damage to property every year (Froude and Petley, 2018). Identifying areas with frequent occurrences of landslides and designating areas with
high landslide probabilities is important to protect human life and economic interest (Colombo et al., 2005; Ludwig et al., 2018). Furthermore, landslides play an important role in the evolution of landscapes by mobilizing and transporting sediment downslope (Moon et al., 2015). Under the generic term 'landslide' a variety of types can be distinguished based on the process and the material involved (Cruden and Varnes, 1996). Several landslide classifications exist that have been refined over the years (Highland and Bobrowsky, 2008; Hungr et al., 2014). When investigating a landslide, gaining knowledge about the
spatial occurrence of landslides can further improve our understanding of the underlying processes causing landslides (Malamud et al., 2004).

The study of landslides reaches from site-specific field investigations to global datasets of landslides and from event-based inspections to long-term monitoring for several years (Alberti et al., 2020; Coe, 2020; Mateos et al., 2020; Svennevig et al.,
2020b). Among the different spatial and temporal approaches of landslide studies, landslide inventory mapping is a common method to investigate the spatial occurrence of landslides (Guzzetti et al., 2012; Galli et al., 2008; Hao et al., 2020). Landslide inventory mapping can be performed remotely, covering large areas, with the option to validate the dataset in the field (Zieher et al., 2016). Traditionally landslide inventories are based on aerial imagery and optical satellite images (Brardinoni et al., 2003; Fiorucci et al., 2011). With the emergence of digital elevation data, the quality and quantity of landslide inventories have
improved substantially (Morgan et al., 2011; Kakavas and Nikolakopoulos, 2021). New areas can be investigated (e.g. forests) and volumes of displaced mass can be calculated (Cavalli and Marchi, 2008). Landslide inventories often contain information about the landslide location, geometry, date of occurrence and damage caused by the landslide (Rosi et al., 2017; Palma et al., 2020).

National elevation mapping efforts and satellite campaigns are extending the areas that are covered by elevation models (Crosby, 2012; Eea, 2016). Advances in sensor technologies and satellite orbit repeat rates are improving the spatial and temporal resolution of the available data, both for optical images and elevation data (*e.g.* Shugar et al., 2021). Remote sensing data provides powerful information for landslide mapping, but a combination of different datasets such as digital elevation models (DEM's) and multispectral satellite images is necessary to overcome the limitations of each individual dataset (Lissak
et al., 2020). The quality of manually mapped landslide inventories strongly depends on the mapping expert's knowledge about the area of investigation (Van Den Eeckhaut et al., 2005). With the emergence of machine and deep learning techniques, landslide inventory mapping is often at least partly performed by computers that identify landslides based on a predefined set of parameters (*e.g.* slope angle, surface roughness) (Chang et al., 2019). The combination of machine learning algorithms and remote sensing data is expected to greatly improve the quality of landslide mapping datasets (Zhong et al., 2020). However,
field validation of these newly created data sets is essential. Semi-automated mapping approaches can reduce the subjectivity introduced by visual landform interpretation (Santangelo et al., 2015). Besides, machine and deep learning algorithms require training data. The limited availability of training datasets and the complexity of automated landslide mapping methods lead to still mostly human efforts in creating landslide inventories (Prakash et al., 2020). Evaluating the quality of landslide inventories is not straightforward and most mapping efforts do not implement quality controls into their inventory (Guzzetti et al., 2012;
Pellicani and Spilotro, 2014; Hao et al., 2020).

Landslide inventories exist on regional, national, international, and global scale (Kirschbaum et al., 2009; Trigila et al., 2010; Damm and Klose, 2015; Herrera et al., 2017). Within Europe, Denmark does not have a national landslide inventory, nor a

legislation framework to incorporate landslides and landslide related damages into national law (Mateos et al., 2020). Landslides are considered a predominant natural hazard in the Nordic countries (Nadim et al., 2008) and a number of case studies investigated landslides in Denmark (Hutchinson, 2002; Prior, 1977). Pedersen et al.(1989) states that Denmark is not a country with a serious landslide problem. However, a recent paper raised concern that the geo-hazard posed by landslides in Denmark is underestimated (Svennevig et al., 2020a).

Previous work suggests that some coastal mudflows and rotational landslides in Denmark are triggered by water infiltration into the soil and underlying Quaternary deposits, activating the shear plane (Prior and Eve, 1975). Water can infiltrate the potential surface of rupture from precipitation events and impermeable clay layers may act as sliding surfaces. Besides, the waves in the coastal areas may remove the toe of landslides and trigger further slope instabilities. Limited records exist of landslides triggered by earthquakes in Sweden, but no records are available for Denmark (Mäntyniemi et al., 2020). Previous

studies have described rockfalls, rotational slides, mudslides, and mudflows, mostly in coastal cliffs in Denmark (Prior, 1977; Pedersen et al., 1989; Busby et al., 2002; Svennevig and Keiding, 2020). With this paper and dataset, we present the first comprehensive landslide inventory for Denmark.

## 2 Study area

Denmark consists of the Jutland peninsula and an archipelago of 394 islands encompassing 43,938 km$^2$ in total with 8,750 km
of coastline (Fig. 1). The landscape is characterized by a low relief with the highest point 171 m above sea level in central Jutland. A long history of agricultural land use has shaped the landscape. Today, around 61% of the area is agriculturally used, 13% are forests, another 13% are transport routes and build up areas, and the remaining land is covered with open habitats and water bodies (Denmark, 2019).

Today's Danish landscape was shaped by numerous glaciations, dominated almost entirely by the two latest, the Saalian, ending c. 130 ka BP and the most recent Weichselian ending c. 16 ka BP, which lead into the Holocene (Houmark-Nielsen, 1999; Houmark-Nielsen, 2011). The current landscape configuration is primarily dominated by the last glacial maximum (LGM) extent reached during the Weichselian at c. 22 ka BP, where a glacial advance from the Northeast reached mid-Jutland, leaving two distinct surface sedimentation regimes: (1) the ice-free west was dominated by sandy glacio-fluvial outwash plains
surrounding older glacial deposits from the Saalian; (2) the ice overridden eastern part of Denmark was dominated by glacial processes depositing tills with a high clay content. The landscape here was mainly shaped during the LGM advance and the numerous re-advances up until c. 16 ka BP (Houmark-Nielsen, 1999; Houmark-Nielsen, 2011). Postglacial isostatic rebound has affected especially the northern part of Denmark, which has been uplifted by up to 13 m relative to the local sea-level, exposing raised beaches and marine terraces.


The Quaternary deposits in Denmark typically comprise tens of meters of glacial diamicton and outwash deposits, but in some places, the Quaternary layer is thinner. Generally, the distribution of Pre-Quaternary deposits follows a pattern from older to younger from the northeast to the southwest (Sorgenfrei and Berthelsen, 1954). The oldest Pre-Quaternary deposits are found on the island of Bornholm to the southeast where a Precambrian crystalline basement and Palaeozoic and Mesozoic sediments
are exposed. In northernmost Jutland, Lower Cretaceous sediments are found below thick Quaternary deposits. Sediments are progressively younger to the southwest and in southern Jutland Miocene and Pliocene clastic sediments are found. Of particular relevance to landslide occurrence are Paleogene plastic clays (Prior, 1977). These are present in a belt from the Limfjorden area in the northwest through eastern Jutland and northwest Zealand and are in many places found inter-thrusted with glaciogenic deposits in glaciotectonic complexes. Cretaceous and Danian limestone cliffs are found locally in a belt northeast





of the Paleogene plastic clays but are most prominent on Zealand at Stevns Klint and on Møn at Møns Klint where rock falls

        occur (Busby et al., 2002; Hutchinson, 2002; Pedersen and Møller, 2004; Pedersen and Gravesen, 2009; Pedersen and Damholt,

        2012).


**Figure 1. Landslide inventory of Denmark. Red dots show 3202 mapped landslides. Dashed line indicates the maximum advance of the ice sheet during the Weichsel glaciation (Houmark-Nielsen, 2011). Place names mentioned in the text along with positions of panels in Fig. 2 are shown.**



Open waters occur in many places in Denmark (Fig. 1) and the glacial landscape is often eroded along its fringes by coastal
processes. Waves induce large swash run up on the beaches and cause erosion of the glacial landscape forming coastal cliffs.
These relatively steep cliffs are susceptible to landslides, if the conditioning geology is present. The landslides in the coastal
cliffs are presumably sensitive to a combination of water infiltration and specific run-off patterns over impermeable layers in
the substrate, and to wave erosion of the cliff toe by swash run up during high water levels under storm conditions (Schou,
1949). The eroded sediment of the coastal cliffs and specifically the landslides in the cliffs, are further transported towards
deeper water in a cross-shore direction or along the shores by wave-driven longshore currents forming accreted forms like
barrier islands and spits (Kabuth et al., 2013; Kabuth and Kroon, 2014).

## 3 Methodology

### 3.1 Data sources

The main datasets used in this study are a high-resolution DEM from 2015 and orthophotos provided by the Danish Agency
for Data Supply and Efficiency (SDFE). The national DEM is produced from airborne LiDAR scans with a spatial resolution
of 40 cm and is freely available (Geodatastyrelsen, 2015b). Several multi-temporal nationwide orthophotos with a resolution
of 12.5 cm complement the mapping effort for visual validation of landslide features in the landscape (Geodatastyrelsen,
2015a). Table 1 shows a complete list of the datasets used to map landslides in this study.

**Table 1. Freely available data from the Danish Agency for Data Supply and Efficiency (SDFE) used in the landslide mapping. See Data availability section for links to the datasets. Adapted from Svennevig et al. (2020a).**

| Name | Type | Year | Source | Resolution (cm) |
|---|---|---|---|---|
| Geodanmark 2020 | Orthophoto | 2020 | SDFE | 12.5 |
| Geodanmark 2019 | Orthophoto | 2019 | SDFE | 12.5 |
| Geodanmark 2018 | Orthophoto | 2018 | SDFE | 12.5 |
| Geodanmark 2017 | Orthophoto | 2017 | SDFE | 12.5 |
| Geodanmark 2016 | Orthophoto | 2016 | SDFE | 12.5 |
| Geodanmark 2015 | Orthophoto | 2015 | SDFE | 12.5 |
| Denmark's Elevation Model | DEM | 2015 | SDFE | 40 |
| DDOland2014 | Orthophoto | 2014 | SDFE | 12 |

### 3.2 Landslide Mapping

A detailed description of the method is given in Svennevig et al. (2020a). The nationwide freely available 40 cm resolution
DEM from 2015 is visualized as a multidirectional hillshade model. Landslides are mapped based on their morphological
expression in the multidirectional hillshade model when a scarp and a displaced unit are observed (Fig. 2). Mapped landslides
are classified into coastal (< 300 m to the coast) or inland (> 300 m from the coast) landslides and categorized by their type of
movement (fall, slide, flow spread) following the classification from Hungr et al. (2014). Several 12.5 cm resolution
orthophotos annually from 2014-2019 are supporting the investigation (Table 1). The method applied here is similar to
Svennevig (2019) and simplified from Slaughter et al. (2017) and Burns & Madin (2009).

### 3.3 Quality control

Two experts mapped landslides in about half of Denmark each. After completion of the initial mapping, a verification of the
mapped polygons was performed by the other expert. Afterwards, an additional validation of the landslide inventory created

by the two experts was performed by having a third expert mapping landslides in various randomly selected subsample areas
to evaluate the completeness of the inventory and to estimate the bias of the mapping first two experts. To achieve this, the

area of investigation was subdivided into 658 tiles with a size of 10 x 10 km. Out of the 658 tiles 192 tiles were randomly
selected, creating a subsample with a confidence level of 90% and an error of 5% (Fig. 3).

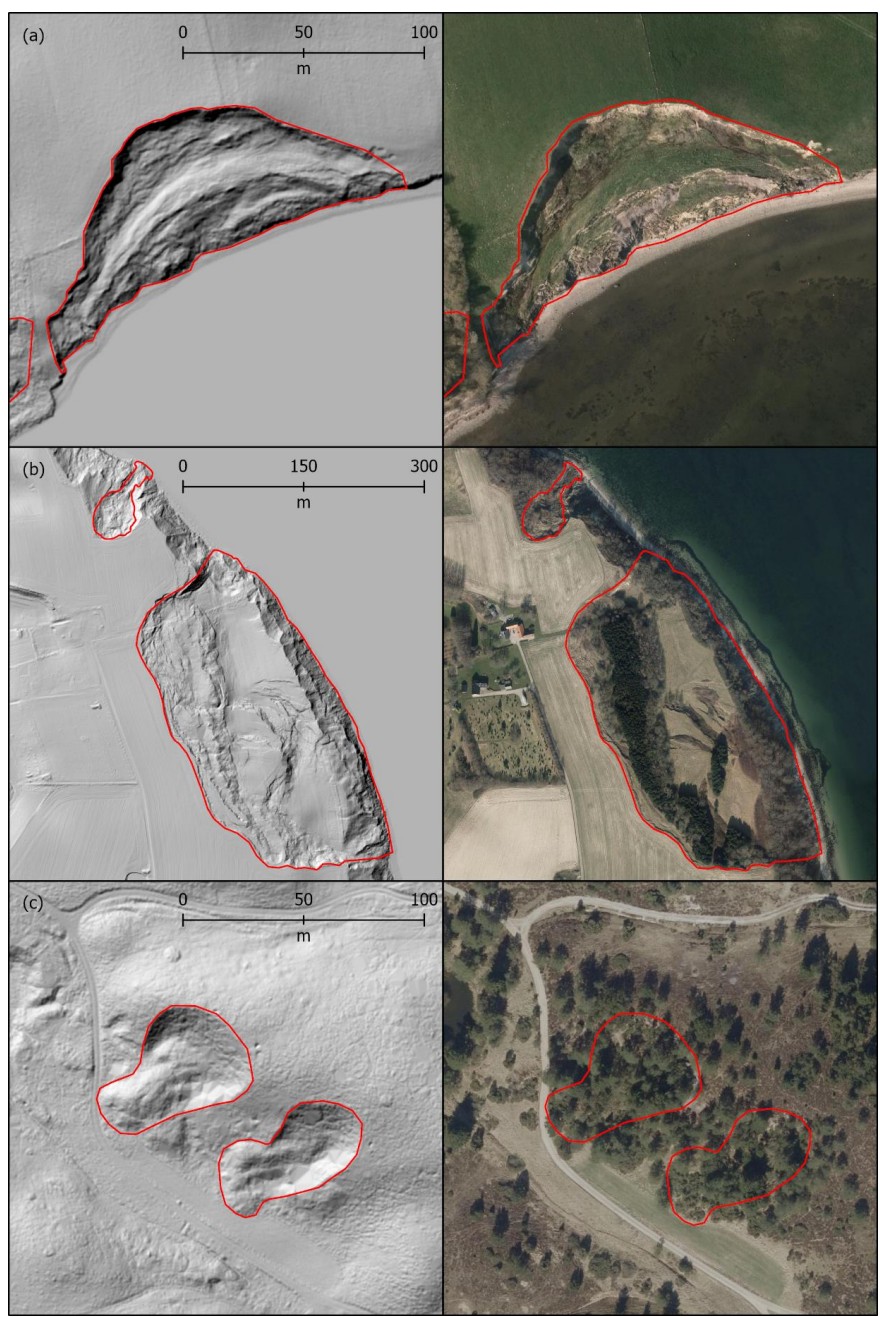

**Figure 2.** Examples of mapped landslide polygons in the hillshade model (left) and Orthophoto from 2015 (right) (Geodatastyrelsen,

2015a, b). Shallow coastal slide (a), coastal flow and deep-seated slide partly obscured by agricultural land use (b) and two shallow
inland slides visible in the hillshade model but covered by vegetation in the Orthophoto (c).



## 4 The landslide inventory

The landslide inventory consists of 3202 unique polygons of mapped landslides. The count of types of movement and the number of coastal and inland landslides are shown in Table 2. Alongside the polygonal shape, every landslide is associated

with a unique identifier. The area ($m^2$) and perimeter length (m) of every landslide are provided as well as the X & Y coordinates of the center point. By area, the largest slides comprise 327,001 $m^2$. Landslides were mapped to a minimum area of 25 $m^2$. Analysis of the mapped landslides shows that most landslides in Denmark are shallow rotational slides. However, the database underrepresents processes with undistinguishable morphologies and expressions in the DEM such as rockfalls and mudflows. However, there are only a few areas in Denmark with the geological preconditions facilitating rockfalls. The

vast majority of landslides recorded are located in landscapes covered in glacial till. Although all mapped landslides must have occurred after the last glaciation, as their morphological expression would have been erased by the activity of the ice sheet, there is no data available in the landslide database when individual landslides emerged. Landscapes that were not covered by ice during the LGM are almost entirely absent of landslides today (Fig. 1).

**Table 2. Landslide types of movement and setting**

| Type of Movement | Coast | Inland | Total |
|---|---|---|---|
| Fall | 62 | 0 | 62 |
| Slides | 2488 | 335 | 2823 |
| Spreads | 1 | 115 | 116 |
| Flows | 155 | 46 | 201 |
| **Total** | **2706** | **496** | **3202** |

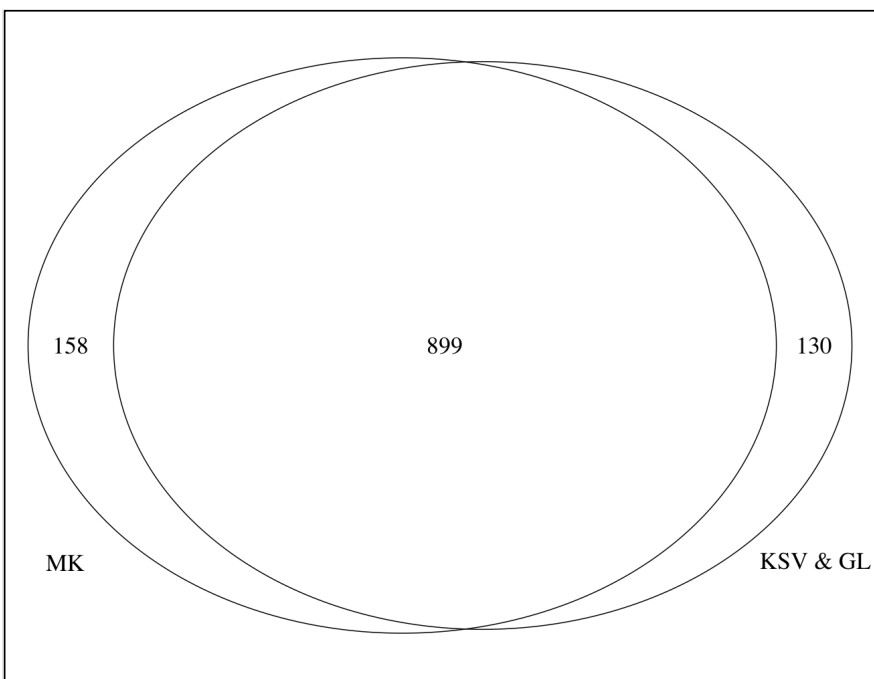

**Figure 3. Venn diagram with the number of mapped landsides in the randomly selected tiles by the two initial experts and the quality control (MK, KSV & GL: 899), the number of landslides only mapped by the quality control (MK: 158) and the number of landslides**

**mapped only by the two initial experts (KSV & GL: 130).**

In most cases, the mapped landslides record single events with process durations that span from an instantaneous event to several decades or even centuries and thus some are still active while others are inactive landforms today. Therefore, the present landslide inventory only represents a snapshot of the landslide activity in Denmark at the time of recording from the 2015 DEM. However, the landslide inventory does not contain any information about current or past activity or inactivity. In

some cases, landslide areas overlap each other making it more difficult to distinguish individual landslide morphologies. Without dating every single landslide, a further distinction is not possible in these cases. Land use such as farming and infrastructure development may have led to an underrepresentation of landslides in these areas due to intensive cultivation and site development, especially in the inland areas. Nevertheless, around 85% of the mapped landslides are in coastal environments, often on a cliff at the edge of agriculturally used land. Farmers usually avoid those steep slopes with their heavy

and expensive equipment. In some areas along the coastal cliffs, abandoned quarries show morphological expressions similar to landslides in the DEM. The absence of landslide deposits can be the only distinction between the morphological expression of a coastal quarry and a landslide in the DEM. Occasionally quarries may have been mistakenly mapped as landslides during the mapping. In some cases, landslides evolved on the steep slopes of a quarry, sliding into the former pit and in other cases, quarry activity may have overprinted landslides.


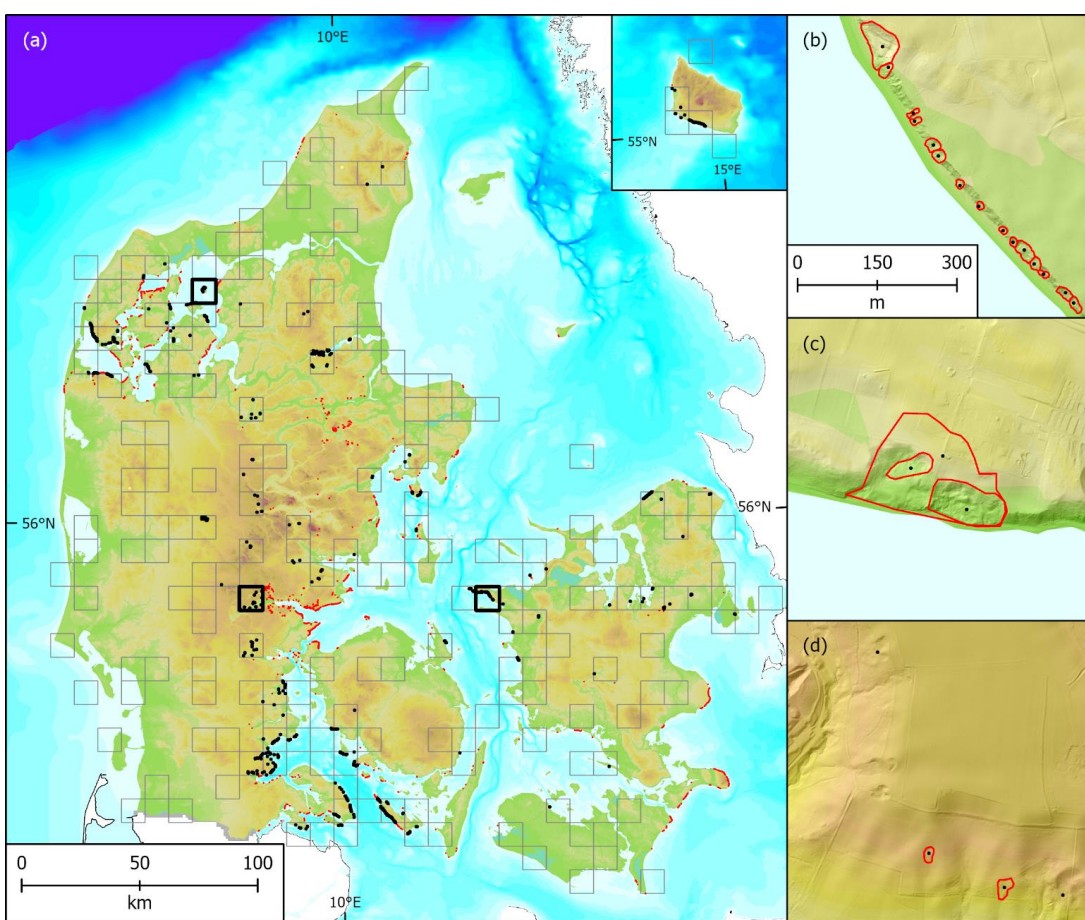

**Figure 4. Landslide inventory quality control with 192 randomly selected tiles across Denmark. Black dots show the 1057 landslides mapped by the third mapper (a), sequence of mapped landslide polygons along the coast with a high accordance of quality control points (b), nested landslides with quality control points for each polygon (c) and mapped inland landslides with quality control points**

**that show additional landslides that were missed by the initial mappers (d).**



Within the area of the subsample plots for quality control, the two experts had initially mapped 1029 landslides and the quality control mapped 1057 landslides, a difference of 2.7%. However, 899 of those landslides were identical, 130 landslides were only mapped during the initial mapping and 158 only during the quality control (Fig. 3). Provided that the combined landslide mapping effort of the initial investigation and the quality control detected the true number of landslides (1187), the initial effort

discovered 87% and the quality control 89% of all landslides. Furthermore, 142 (4.2%) landslides in the entire study area were validated by visiting the landslides in the field or by mentions in other resources such as previous publications or newspaper articles.

Based on the careful observation of the entire study area and the implemented quality control, the landslide inventory can be

considered 87% complete with a confidence level of 90% and an error of 5% for the 2015 DEM. However, a few landslides always remain undetected and new landslides will have emerged since the DEM was recorded. According to the landslide inventory protocol from Burns & Madin (2009) we only mapped landslides with a moderate to high confidence. The high confidence level in combination with the high quality of the input datasets, lets us conclude that all landslides included in the database are actually landslides.

**5 Data availability**

The landslide dataset and a document with metadata are freely available from https://doi.org/10.6084/m9.figshare.16965439.v1 (Svennevig and Luetzenburg, 2021) and can also be viewed through a web map environment (https://data.geus.dk/landskred/) where layers such as the hillshade model, soil map, Pre-Quaternary geology, etc. can be displayed for context. The landslide dataset is provided in the form of an Environmental Systems Research

Institute (ESRI) shapefile including the following attributes: landslide ID, area, perimeter length, center point coordinates, coastal or inland and movement type. The definitions of each attribute are provided in an additional metadata text document. The DEM is available for download in 10 km tiles (https://download.kortforsyningen.dk/content/dhmterræn-04-m-grid).

**6 Significance of the dataset**

The motivation for creating and freely providing this landslide inventory is twofold:


1. The first national landslide inventory for Denmark is an important step towards a more comprehensive hazard and risk framework for Denmark. The inventory enables local, regional and national stakeholders to implement landslides into their risk reduction strategies. Furthermore, a legislative framework implementing landslide risk and damage may build upon this dataset. With the expected increase in global landslide activities due to climate change, a landslide risk reduction strategy is

now more important than ever before (Gariano and Guzzetti, 2016). In Denmark, a combination of increases in frequency and magnitude of heavy precipitation events, ground water level rises, storm surges and a general increase in relative sea level make a higher landslide activity in the future very likely. Therefore, it is crucial to better understand the underlying processes causing landslides and develop effective risk reduction strategies to protect human lives and property.

2. Providing an expert based high quality, scientifically evaluated landslide inventory to the machine and deep learning research community. The landslide data set is validated and extends the availability of urgently needed training datasets for automated mapping methods. The consistently high amount of time required to manually compile landslide inventories stands in contrast to the increase in data available for landslide mapping. Future challenges in landslide inventory mapping lie in developing methods to reliably automate the process. The present dataset provides a valuable resource to train and develop future





algorithms for this task. Additionally, this is one of the few landslide inventories providing a statistical error estimation of the completeness of the number of mapped landslides.

## 7 Author contribution

KSV: conceptualization. GL & KSV: data curation, formal analysis, visualization & writing original draft. MK: quality control. All: discussion of dataset, review & editing.

## 8 Acknowledgements

GL has received funding from the European Union's Horizon 2020 research and innovation program under the Marie Skłodowska-Curie grant agreement No 801199.

## 9 Competing interests

The authors declare that they have no conflict of interest.

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
