# Peer review of "A national landslide inventory of Denmark"

_Earth System Science Data, 2021_

## Author Response (AR1)

**Reviewer 1**

The article presents relevant new data in the form of a national landslide inventory for Denmark. The new data collected and presented in the article fills a gap in the European and international panorama of landslide mapping efforts and related resources. I commend the authors for their efforts, and for making the collected data publicly available.

Overall, the landslide data collected and presented in the article are of potential scientific interest to a broad community, and are of possible practical use. The landslide data appear to be collected, treated, and organised in a manner scientifically and technically consistent with common practices to produce national-scale landslide inventories. Therefore, it is my opinion that the work deserves publication, pending some improvement concerning additional information on how the inventory was produced, the presentation of the landslide data, the text of the article that needs adjustments, and the references.

I have a few general remarks and some specific comments. I list them below, in the hope that they will help the authors improve their already interesting article.

We thank the reviewer for their valuable comments and the positive feedback on the quality of our work. Detailed point-by-point responses can be found below.

**General remarks**

1.  Denmark is a European Country with vast oversee territories, including the Faroe Islands and Greenland, in the Atlantic Ocean. For readers – like me – non entirely familiar with the political and physical geography of Denmark, the authors should explain clearly what part the Country is covered by the new inventory.

    You are right that the political geography of Denmark can be complicated - however, Denmark as a sovereign nation is part of the Kingdom of Denmark, which also includes the constituent countries Greenland, and the Faroe Islands. The term Denmark refers only to the "southern" part of the Kingdom of Denmark.  The geographical extend of Denmark is also shown In Fig. 1, which is referenced in the same sentence.

2.  Inspection of Figure 2 reveals that the authors have used a single polygon to encompass all parts of a single landslide, including the source area and the landslide deposit. The authors should explain why they have adopted this strategy to map the landslides, including e.g., the time available to complete the mapping, the insufficient resolution and vertical accuracy of the DEM, the difficulty in separating the source area from the main deposit, systematically. To some extent, the choice made not to separate different landslide parts may limit the possibility of using the inventory to calculate the volume of the individual landslides and to infer erosion rates.

    The main reason for not distinguishing between surface of rupture and displaced mass of a landslide is the high number of coastal landslides. Coastal erosion limits the possibility to separate the source area from the main deposit. The toes of the coastal landslides are often eroded by waves and not visible in the DEM anymore. Separately mapping area of erosion and area of deposition would have added a big bias to the inventory. To make this clear in the text, it now reads: 'Coastal erosion makes it difficult to separate the source area from the main deposit and the landslide foot is often partly removed by wave erosion. Therefore, landslides are mapped in a single polygon and the mapping did

not distinguish between source area and landslide deposit.' We roughly estimate the compilation of the landslide inventory cost about eight weeks of mapping and data management.

The quality control took about one week. The mapping effort started as a side project with focus on some specific areas before we decided to map all of Denmark. Therefore, we cannot give a reliable estimate of mapping time per unit area or landslide.

3. A related item has to do with the mapping of landslides partially or totally inside larger landslides. The authors should explain how they have addressed the problem. I have downloaded and inspected the shape file for the inventory, and found that overlapping landslides are treated as separate and independent polygons. This complicates the analysis of the inventory. As an example, summing the areas of the individual polygons will overestimate the total area affected by landslides in the study area – as it will count the overlapping areas twice, at least.

We added information about the mapping of overlapping landslides in the method section and it now reads: 'Subsequent landslides in the same area are mapped as overlapping independent polygons when it was possible to clearly differentiate between varying morphological features.'

4. The inspection of the inventory in a GIS revealed long girdles of landslide polygons, each representing individual landslides, mainly along the coasts. This is a common geomorphological setting where landslides erode and modify a "mesa" like, or "table top" morphology, the result of a nearly horizontal layering of rocks of different mechanical characteristics. A problem when mapping landslides in this setting is the lateral separation of the individual landslide blocks, which may not be trivial and it may be subject to human interpretation. I recommend that the authors address the issue, and show at least one example of their landslide girdles and the accuracy of their mapping in these areas.

We added a panel in figure 2 to show an example of a sequence of coastal landslides. Furthermore, we mentioned these landslide sequences in sub-section 3.2 and it now reads: 'Along the coast, landslide morphologies occurred in sequences next to each other. When it was not possible to separate single landslides in the hillshade model, succession rates of the vegetation, visible in the orthophotos, were used to distinguish between morphologies. '

5. In sub-section 3.2, the authors provide information on how the mapping was performed by two experts, with a third expert performing an unsystematic verification of parts of the inventory. However, how this crucial part of the preparation of the national landslide inventory was performed is not sufficiently clear. The authors should show (e.g., in Figure 1) which tiles were mapped first by one expert, and which by the other expert (KSV, GL). This may outline a source of potential geographical biases that may be present in the national inventory. Second, the authors should explain what kind of verification was performed by the second expert. Was it a completely independent survey, or the second expert had access to the map of the first expert? Where disagreements emerged, how were they resolved? Similar questions arise for the validation performed by the third expert. Were the independent mappings of the two (first and second) experts available to the third expert, or only the joint (verified) result of both experts, or none of them? Again, where a disagreement emerged, how was it resolved? Was the final mapping changed based on the opinion of the third expert, or based on some form of mutual agreement? In the latter case, how this was accomplished? In general, what was the subject of the second and the third mappings. Did the second and the third experts check only the existing mapping and refined it e.g., changing the geometry of

the polygons representing the landslides? Did they change the classification of the landslides? Did they added or deleted landslide polygons identified by the first (or the first and the second) expert? These are important issues that influence the quality, and hence the usability of the landslide dataset. I recommend that the authors address these issues, albeit briefly.

We added information regarding the verification and quality control methodology in chapter 3.3. The subset area mapper 3 validated is shown in figure 4. Mapping Expert 1 & 2 verified each other's mapping effort including geometry, disagreements were marked and resolved mutually. Mapping expert 3 performed an independent validation of the mapping from mappers 1&2 in a subset of the study area. Classification and geometry were verified by mappers 1&2 whereas mapper 3 only validated the detection of a landslides in the subset to quantify the completeness of the inventory. Landslides that were only detected by mapper 3 during the validation of the subset were evaluated by mapper 1&2 after estimating the completeness of the original database. We added more information in chapter 3.3 and it now reads: 'The third mapper used the same datasets and applied the same criteria for mapping a landslides like the two initial mappers, but had no knowledge about the already mapped landslides in the subset area. The quality control mapper mapped landslide points and an agreement between the two initial mappers and the third mapper was reached, when the quality control point fell within the initial landslide polygon. After estimating the completeness of the inventory based on the comparison of the two independent mappings, landslides that were detected by the third mapper, but not the first tow mappers were added to the inventory. However, in some cases the first two mappers did not agree with third mapper and not all landslides were added to the final database.'

6.  The authors should provide information on the time and human effort required to prepare the national inventory, including an account or an estimate of the overall time and of the persons / month or persons / year effort required to prepare the inventory, and for the single steps of the work e.g., gathering and organizing the DEM and preparing the hillshades, gathering and organizing the orthophotographs, the visual inspection and digital mapping of the landslides, the validation of the mapping, the open publication of the results.

    The data sources (DEM & Orthophotos) are provided through a WMTS service by the Danish Agency for Data Supply and Efficiency (SDFE). Unfortunately, we did not properly recorded the time required for the mapping nor the quality control and open publication of the results.

7.  As a last general remark, I recommend that the authors check the citations they have selected. I am not convinced that all of them are fully appropriate, at least in the locations they are used in the text. Also, I encourage the authors to look for additional, mostly recent references on landslide detection and mapping methods and tolls.

    Thank you for your remark. To our knowledge, all citations are appropriate. We would appreciate more specific comments which citations could be approved.

**Specific comments**

**Title**

8. I suggest the author consider the slightly different title "A national landslide inventory for Denmark", or "National landslide inventory for Denmark".

We changed the title accordingly.

**Abstract**

9. The abstract should focus strictly on the main topic of the article i.e., the new national landslide inventory for Denmark. The abstract should address the content of the inventory, focusing on the method used to compile, validate, organize, and publish openly the new landslide information. Any other consideration on the global relevance of the landslide problem, the risk it poses, and the potential (future, and therefore currently not proven) use of the landslide information for different scopes, should be removed from the abstract.

We deleted the future outlook in the outlook.

**Introduction**

10. In the journal aims & scope (https://www.earth-system-science-data.net/about/aims_and_scope.html) one reads that "Articles in the data section may pertain to the planning, instrumentation, and execution of experiments or collection of data. Any interpretation of data is outside the scope of regular articles". It follows that most of the text in the Introduction is out of scope for the journal. I understand the need, and I appreciate the attempt the authors have made to frame their work in a broader perspective, but the Introduction is too long and not focused on what should be the main scope of the paper: presenting a new, valuable, national landslide inventory for Denmark. I recommend that the authors reconsider the text in the Introduction, reducing it considerably, and focusing it on the main scope of the paper

We substantially shortened the introduction, focusing on the main scope of the paper.

**Study area**

11. Following up on the same argument made before for the Introduction, the description of the Danish landscape and the recent geological history of Denmark is probably out of scope for an article in this journal, unless the information was instrumental to the compilation of the landslide inventory. I encourage the authors to consider the point, and change the text accordingly.

We removed information about the underlying Geology but kept the description of the land surface since we think this is substantial for the compilation of the landslide dataset.

12. In Figure 1, the authors use colours to show surface and submarine terrain elevation, but they do not provide a legend for the colours used. The dashed grey line shows the maximum advance of the ice sheet during the Weichsel glaciation. However, it took me a while to understand what side of the line was covered, and what side was not covered by the ice. For the readers who are unfamiliar with the geography of the region, it would be good to show the boundary using a line with a different, asymmetric symbol. The acronym LGM is not explained in the Figure caption. The locations of Fig. 2a, 2b, 2c, are not easy to spot, at first sight. The authors should consider using a larger font, bold

characters, or a different text colour. Figure 2b is similar (albeit not identical) to Fig. 2a in Svennevig et al., 2020, GEUS Bulletin 44, 5302. This should be clarified in the Figure, by writing e.g., "modified from", or the like.

We clarified the acronym LGM in the figure caption as well as which part of Denmark was covered under the ice sheet during the last glaciation. A further description of the geography of Denmark is provided in chapter 2.

**Methodology**

13. In line 127, the authors write "visual validation of landslide features in the landscape". What does it mean, precisely?

    We mapped the landslides based on their morphological expression in the hillshade model and validated this by crosschecking with the orthophotos. Most of the times, landslide morphologies are visible in the DEM and the orthophotos, sometimes only in one of the two datasets.

14. Sub-section 3.2, Landslide mapping, is too concise. I understand the authors point the reader to the – freely available – work by Svennevig et al. (https://doi.org/10.34194/geusb.v44.5302); but some description on the method and tools used to collected the landslide data is important to assess the quality of the data, and to decide on the use of the data. I recommend that the authors expand this sub-section.

    We expanded this section with additional information regarding the method. Additionally to the already existing description of the method it now reads: 'The identification of a landslide in the multidirectional hillshade model is supported by additional morphological features such as a crown, transverse cracks, main body or foot in many cases. Coastal erosion makes it difficult to separate the source area from the main deposit and the landslide foot is often partly removed by wave erosion. Therefore, landslides are mapped in a single polygon and the mapping did not distinguish between source area and landslide deposit. Subsequent landslides in the same area are mapped as overlapping independent polygons when it was possible to clearly differentiate between varying morphological expressions. Along the coast, landslide morphologies occurred in sequences next to each other. When it was not possible to separate single landslides in the hillshade model, succession rates of the vegetation, visible in the orthophotos, were used to distinguish between morphologies.   Landslides that originate from before the last glaciation are not included in the database due to the high uncertainty of the morphological expression in the DEM.'

15. Sub-section 3.3, Quality control is important and interesting, but it also too concise. See my general remark on this topic.

    We added information. Please see our answer of the general remark.

**The landslide inventory**

16. The separation of landslides into "coastal" (or "coast") and "inland" landslides is not fully clear. Do coastal landslides have (currently) their toe in the sea? Or are coastal landslides slope failures that affect slopes that have (currently) their toe in the sea? Are "inland" landslides at a minimum distance

to the sea? Are landslides on the slope of a lagoon or lake (if any) classified as "coastal" landslides? In general, I recommend that the authors explain why they have made the separation between "coastal" and "inland" landslides, and that they provide a clear definition for a "coastal" landslide and for an "inland" landslide.

The classification of landslides into coastal and inland is only based on the distance of a landslide to the shore. Landslides that are closer than 300 m to the shore are classified coastal and landslides that are more than 300 m from the coast are classified inland. In the method section it reads: 'Mapped landslides are classified into coastal (< 300 m to the shore) or inland (> 300 m from the shore) landslides and categorized by their type of movement (fall, slide, flow spread) following the classification from Hungr et al. (2014).'

17. In line 155, the authors specify that the area (m2) and perimeter length (m) are given in the inventory. They should specify that these are planar figures i.e., they represent the area and perimeter of the landslide as show in the map, and not their true area and perimeter length in the field. The latter are larger and longer, given the fact that the landslides form and develop in a sloping terrain. It would be good if the authors could calculate and provide this additional information with their inventory i.e., with the data ad metadata in the shape file. This would allow users to analyse the inventory without having to download the DEM from which the inventory was obtained.

We specified that it is the planar area and perimeter length. Due to the size and data structure of the DEM we cannot calculate the true area and perimeter length ourselves. It now reads: 'The planar area (m$^2$) and perimeter length (m) of every landslide are provided as well as the X & Y coordinates of the center point.'

18. In line 156, the authors give the area of the largest mapped landslide as 327,001 m2. Clearly the one m2 is somewhat "fictitious", in the sense that a very small change in the mapping may have resulted in a different total area for the landslide. Given the fact that the authors have not mapped landslides with a (planimetric) area smaller than 25 m2, I recommend that the authors present their data to the nearest 25 m2.

Yes, you are right. We have changed the numbers accordingly.

19. In line 171, the authors write "In most cases, the mapped landslides record single events with process durations that span from an instantaneous event to several decades or even centuries and thus some are still active while others are inactive landforms today". The sentence appears contradictory, and needs some clarification. I recommend that the authors provide a clear definition of a what they consider an "event", or a "landslide event". A "process duration" of several decades or centuries implies that the same landslide has been active for several decades or centuries; or not?

We interpret most of the mapped landslides as single events, however, the duration of each event may have varied from instantaneous to several decades or centuries. If some of the mapped landslides are clearly not a single event, then that could be described.

We clarified the sentence and it now reads: 'We interpret most of the mapped landslides as single events with process durations that span from an instantaneous event to several decades or even centuries and thus some are still active while others are inactive landforms today. Landslides that are clearly not a single event are mapped as separate polygons.'

20. In lines 199 and 200, the authors write "Based on the careful observation of the entire study area and the implemented quality control, the landslide inventory can be considered 87% complete with a confidence level of 90% and an error of 5% for the 2015 DEM." These are clear and important figures given. I am sorry, but from the previous discussion and the presentation of the inventory, I do not understand how the figures were calculated, or estimated. What does it mean that the inventory is 87% complete? That it misses 13% of the total number (or area) of the landslides? And what landslides? All the landslides that are (in principle) visible in the DEM and the orthophoto maps used for the mapping, and that for whatever reason where not detected and mapped? Or all the landslides that have occurred in the study area since the last glaciation? The difference may be significant, and can possibly be sized from frequency-density plots of the landslide areas (see e.g., Malamud et al., 2014). How was the 90% confidence level calculated, and what does it mean, precisely. Ultimately, how the 5% error for the 2015 DEM was calculated, and how this apparently small error has affected the visual detection of the landslides from the hillshades?

We divided the area of investigation (Denmark) into 658 10x10 km tiles and randomly selected a subset of 192 tiles which corresponds to a confidence level of 90% with and error of 5%. In those 192 tiles a third mapper performed an independent point mapping of landslide locations following the same methodology. Within the area of the subset, 899 landslides where mapped identical, 158 landslides were only mapped by the third mapper and 130 landslides were only mapped by the two initial mappers. Provided that the combined landslide mapping effort of the two initial mappers and the third mapper detected the true number of landslides (1187), the initial effort discovered 87% and the third mapper 89% of all landslides. We estimate that we missed 13% of the total number of landslides in the DEM. Please find more information in our answer of the general remark.

**Significance of the dataset**

In this section, I am not convinced the authors do justice to their important work.

21. The authors provide two main motivations for the work. The first is a step towards "a more comprehensive hazard and risk framework for Denmark". This may be the case, but before embarking in a comprehensive hazard and risk framework for Denmark, it is plausible that the data can be used by landslide scientists and practitioners to construct landslide susceptibility and hazard models for Denmark. Discussing the same motivation, the authors suggest that a use of the new dataset is to "develop effective risk reduction strategies to protect human lives and property". Again, this may be the case, but it is not clear the extent to which landslides in Denmark threaten human lives and property. Since landslide activity depends on climate, it seems to me that a potential use of the dataset will be investigating and monitoring the effects of the changing climate along the high coasts of Denmark, confronting them with the similar effects on the "inland" landslides.

We agree the dataset itself needs to be implement into hazard and risk frameworks or risk reduction strategies. We clarified that making the dataset available empowers everyone with an interest to do so.

22. The second motivation is to provide landslide information "to the machine and deep learning research community." Although I recognize the scope and potential of AI-based methods in several fields of science, including landslide modelling and landslide hazard and risk assessment, I would not limit the use of this new national landslide inventory to the machine and deep learning research community. Several other promising research can be attempted by exploiting this dataset that do not require AI, including machine and deep learning, for modelling and predictions.

    We agree, the landslide inventory can be used by more scientists than 'only' the machine and deep learning community. We clarified that in the text.

**Reviewer 2**

The paper presents the first national landslide inventory for the country of Denmark, generated through visual-image interpretation of hill-shading images from a LiDAR survey, with additional high-resolution optical images. The authors have also made an evaluation of the accuracy of their mapping by checking each other's inventories, and by the checking of an independent third mapper. The resulting landslide inventory is available as a shapefile or in a web browser.

Thank you very much for reading the manuscript and for the positive feedback.

23. The dataset is very interesting and the introduction paper is certainty worth publishing, even though there is already another paper that describes the mapping as well (Svennevig et al., 2020a). In that paper, you indicated also different attributes in the landslide inventory database (shape, proximity to coastline, morphology indicator of recent activity, given name, name of mapper, hazard potential). Why are these not included in this version?

    We include information about shape (polygon), type of movement (slide, fall etc.), coastal or inland, center coordinates, planar area and perimeter length in the database (https://doi.org/10.6084/m9.figshare.16965439.v1). We only provide landslides that occurred after the last glaciation (recent) but we do not have any information about more recent activity. We provided information about the mapper and the quality control in the database. We only have local names for a very small number of landslides.

24. It would be advisable to indicate in the landslide inventory, which landslides have been confirmed by an independent mapper (by adding a field in the database with 1 and 0), which landslides have been field checked (by adding a field in the database with 1 and 0), and which landslides have been reported in other publication (by adding a field with the DOI of the publication).

    We added information to the database regarding field validation and confirmation by the independent mapper. See also the metadata file of the database.

25. It would be interesting to provide a more worked out analysis of the landslide inventory, based on the area/frequency of the various types of landslides.

    Yes, this could be part of an original research paper in a scientific journal, but would exceed the scope of this data description paper.

26. The usefulness of the high-resolution optical images in mapping landslides, as compared to the hills shading images from the LiDAR DSM could be discussed more. From the examples shown it appears that the latter are much more useful for landslide inventory mapping.

    We agree that many landslide morphologies are much better visible in the LiDAR DEM. Therefore, we used the LiDAR DEM as our main data source for the landslide mapping. However, we did not quantify how many landslide where only or better visible in which dataset.

27. The quality control procedure that you applied is very nice. However, the Venn diagram in Figure 3 is a bit confusing. I would expect three individual circles (of KSV , of GL and of MK). KSV and GL also checked each other's mapping results. Is that not taken into account in the evaluation of the quality?

    KSV and GL checked each other's mapping during the initial mapping, but not independently and we did not use the same zonation for the initial mapping, we used for the quality control. Only the quality control executed by MK was independent and is therefore used to assess the completeness of the inventory.

28. One aspect missing in the paper is a proper estimation of the time involved in landslide inventory mapping. Many papers on automatic mapping claim that manual mapping is too time-consuming. This paper gives an ideal opportunity to quantify the time required, and specific it in time per landslide, and time per unit area.

    We roughly estimate the compilation of the landslide inventory cost about eight weeks of mapping and data management. The quality control took about one week. The mapping effort started as a side project with focus on some specific areas before we decided to map all of Denmark. Therefore, we cannot give a reliable estimate of mapping time per unit area or landslide.

29. The statement on the use of the dataset for deep learning algorithms for landslide mapping is only applicable if these algorithms would be applied using similar quality high-resolution elevation data. It would be good to discuss this further.

    Agreed, and the high quality elevation data is freely available through the Danish Agency for Data Supply and Efficiency (SDFE). It now reads: 'The present dataset provides a valuable resource to train and develop future algorithms for this task. Especially in combination with the freely available DEM, automated mapping methods, can include the elevation data into their investigation. '

30. Even though you mention that you cannot include information on the age or activity of the landslides, based on a data source of a single date, there are certainly indications (also mentioned in Svennevig

et al., 2020) of very ancient landslide complexes that occurred under different climatic conditions. Would it not be useful to include this as an attribute in the database?

We excluded the ancient landslides mentioned in Svennevig et al. 2020 from the published database due to the uncertainty of the morphological expression in the DEM deriving from these very old processes. In chapter 3.2 it now reads: 'Landslides that originate from before the last glaciation are not included in the database due to the high uncertainty of the morphological expression in the DEM.'

31. Is there a procedure to regularly update the landslide inventory?

The landslide inventory will be maintained by the Geological Survey of Denmark and Greenland (GEUS) and that there is no fixed procedure in place to regularly update the database. The actual update frequency of the data base depends on further funding.

**Some detailed comments:**

32. L 40: the statement "new areas can be investigated (forest) " could be improved by adding the functionality of using LiDAR DSM hill-shading images.

We mentioned hill-shading in combination with digital elevation data and it now reads: 'With the emergence of digital elevation data and hill shading those, the quality and quantity of landslide inventories have improved substantially.'

33. L35-36: the statement on the reduction of subjectivity using automated approaches seems naïve to me. When you did the work yourself you have become better at recognizing landslides, and you have a learning curve. Expert interpreters will still function much better than deep learning algorithms.

We agree and deleted the sentence.

34. L 110: Figure 1: Please provide a legend for the elevation. I advise showing the landslides in black so they are different from the elevation colors.

We adapted the figure and landslides are shown in black now.